# Optimizing Outcomes through a Multidisciplinary Team Approach in Endometrial Cancer

**DOI:** 10.3390/healthcare12010064

**Published:** 2023-12-27

**Authors:** Lucia Mangone, Francesco Marinelli, Isabella Bisceglia, Maria Barbara Braghiroli, Valentina Mastrofilippo, Annamaria Pezzarossi, Fortunato Morabito, Lorenzo Aguzzoli, Vincenzo Dario Mandato

**Affiliations:** 1Epidemiology Unit, Azienda USL-IRCCS di Reggio Emilia, 42123 Reggio Emilia, Italy; francesco.marinelli@ausl.re.it (F.M.); isabella.bisceglia@ausl.re.it (I.B.); mariabarbara.braghiroli@ausl.re.it (M.B.B.); annamaria.pezzarossi@ausl.re.it (A.P.); 2Unit of Obstetrics and Gynaecology, Azienda USL—IRCCS di Reggio Emilia, 42123 Reggio Emilia, Italy; valentina.mastrofilippo@ausl.re.it (V.M.); lorenzo.aguzzoli@ausl.re.it (L.A.); vincenzodario.mandato@ausl.re.it (V.D.M.); 3Biotechnology Research Unit, Aprigliano, 87051 Cosenza, Italy; f.morabito53@gmail.com

**Keywords:** endometrial cancer, stage, multidisciplinary team, recurrence, disease-free survival, death, adjuvant treatment

## Abstract

This study aimed to assess the impact of a multidisciplinary team (MDT) approach on outcomes with endometrial cancer (EC) patients, utilizing 2013–2020 data from the Reggio Emilia Cancer Registry. Recurrence rate, treatments, and outcome indicators were compared between the MDT (319 cases) and non-MDT (324 cases) groups. Among 643 cases, 52.4% were over 65 years old, 98% had microscopic confirmation, and 73% were in stage I. Surgery was performed in 89%, with 41% receiving adjuvant therapies. Recurrence rates (10%) were similar between the two groups, but MDT patients who were older and predominantly in stage I exhibited 79% recurrence within one year (21% in the non-MDT group). Disease-free survival (DFS) showed no significant difference [HR 1.1; 95% CI 0.7–1.6], while differences in overall survival (OS) were notable [HR 1.5; 95% CI 1.0–2.4]. The 5-year OS rates were 87% and 79% in the MDT and non-MDT groups. Comparing the 2013–2015 to 2016–2020 study periods, a shift towards caring for older women, more advanced-stage patients, and those residing outside the metropolitan area, along with a greater number of relapsed cases (from 16% to 76%), were accounted for. These findings underscore the impact of an MDT on EC outcomes, highlighting the evolving patient demographics over time.

## 1. Introduction

Endometrial cancer (EC) ranks as the sixth most frequently diagnosed tumor and the thirteenth most lethal cancer worldwide, with over 417,000 new cases and 97,000 annual deaths [1]. Roughly 10,200 new cases of EC, constituting 5.5% of all female cancers, are diagnosed annually in Italy, representing the third most frequent neoplasm among women in the 50–69 age group, with regrettably 3100 deaths each year [2]. In Italy, the incidence trend appears to be slightly decreasing in the period 2003–2014 (APC = −0.1%; −0.4; −0.1), with a similar decrease in mortality from 2003 to 2017 (APC = −0.5%; −2.1%; 1.2) [2]. Despite a favorable prognosis with a remarkable 5-year survival rate of 79% [2], mortality remains high in the case of metastatic disease or recurrence [3]. Several risk factors contribute to the development of EC. Apart from age [4], obesity plays a significant role [5]. Factors influencing hormone levels, such as postmenopausal estrogen use [6], tamoxifen use [7], prolonged years of menstruation [8,9], and even low physical activity and poor diet, elevate the risk of EC [10]. Additionally, individuals with Type 2 diabetes are also at higher risk [11]. Moreover, there is evidence that a family history of EC is associated with an increased risk of EC [12]. Approximately 5% of women diagnosed with EC have a family history of the same cancer, and 2% have a family history of colorectal cancer [13]. Finally, women who have previously had breast cancer or ovarian cancer might also face an elevated risk of developing EC [14]. Conversely, the use of oral contraceptives has been associated with a lower risk of EC [15]. Additionally, carrying a pregnancy to full term is associated with a reduced risk of developing EC later in life [16]. Adenocarcinoma is the most frequent morphology of EC [17]. Notably, since most women with EC experience bleeding at the onset, early diagnosis is reasonably straightforward. As a consequence, 80% of EC cases are confined to the uterus at diagnosis [18]. EC cases are categorized into low, intermediate, intermediate-high, and high-risk groups based on prognostic factors, including the degree of myometrial infiltration and tumor differentiation, tumor size, the presence of vascular emboli and/or lymphatics, lymph node metastases, and histotype [19]. The likelihood of recurrences is more significant in women who are deemed to be at elevated risk [20]. Hence, it is crucial to determine the risk of recurrence and establish an efficient management strategy ab initio for these high-risk patients. Over the past decade, researchers have delved into the molecular profile of EC to more accurately evaluate the recurrence risk [21]. Moreover, in the last three years, this molecular understanding has been introduced into international guidelines, shaping the recommendations for the use of adjuvant therapy [22,23] The introduction of MDT approaches in recent years seems to have changed the dynamics of the clinical behavior of various tumor types. In the case of breast cancer, MDT involvement has shown promising results in improving patient outcomes by influencing treatment modalities, reducing mortality rates, and ultimately increasing overall survival [24]. Furthermore, MDT strategies have demonstrated their potential in reducing the frequency of recurrences of breast cancer [25,26] and even ovarian cancer [27]. Concerning EC, adopting an MDT approach has proven beneficial in enhancing adherence to treatment protocols [28]. Additionally, MDT strategies have shown efficacy in optimizing the overall management of EC patients [29,30].

This study aims to evaluate whether the utilization of a multidisciplinary team can change the outcome for women with EC and to analyze the evolution of MDT implementation over the years.

## 2. Materials and Methods

The study is a single-center retrospective observational study that includes all ECs diagnosed between 2013 and 2020 in Reggio Emilia province. The study includes all malignant infiltrating endometrial cancers diagnosed and incidents in the province of Reggio Emilia in the period under examination; no cases were excluded. The cases were cross-referenced with the local MDT center to verify how many women were followed by the MDT. The MDT panel meets once a week and includes pathologists, oncologists, gynecologists, radiotherapists, and psychologists for integrated management of the patient.

EC cases were classified as topography C54 in the International Classification of Diseases for Oncology, Third Edition (ICD-O-3) [31]. The Reggio Emilia Cancer Registry (RE-CR) relies on three primary information sources: pathology reports, hospital discharge records, and mortality data, which are integrated with laboratory tests, diagnostic reports, and information from general practitioners. Consulting hospital medical records yielded information on stage (TNM 8th edition) [32], as well as surgery, chemotherapy, and radiotherapy. All information was obtained from the Cancer Registry except for the variables on stage, surgery, chemotherapy, and recurrence, which were retrieved from medical records since they are not routinely recalled from CRs.

The RE-CR covers a population of 532,000 inhabitants and is considered a high-quality CR due to its current extension until the end of 2020, a high percentage of microscopic confirmation (98.8% for EC), and a very low number of Death Certificate Only (DCO < 0.1%) [33]. The Cancer Registry collects data and information from current flows to generate incidence, mortality, prevalence, and survival statistics for the resident population and demographic subgroups as required by the epidemiological report, defined by Law No. 29 of 22 March 2019, that regulates the cancer registries in Italy. The Law exempts the registries from collecting informed consent. The procedures for conducting epidemiological analyses of the Reggio Emilia Cancer Registry data were approved by the provincial Ethics Committee of Reggio Emilia (Protocol No. 2014/0019740 of 4 August 2014).

The descriptive statistics were calculated for age at diagnosis (classified into three age groups: <50, 50–65, and over 65 years), period of diagnosis (two groups: 2013–2015 and 2016–2020), method of diagnosis (histological or clinical/instrumental), stage, therapies, district, and recurrence (excluded stage IV) stratified by group status (MDT vs. non-MDT) and period of diagnosis. We decided to eliminate stage IV from the recurrence calculation to emphasize the appearance of any loco-regional recurrences or distant metastases. A multivariable Cox proportional hazard regression model was constructed to investigate the association between stage, age, therapies, MDT, recurrence, and overall and disease-free survival (time was expressed in years). The time-to-event outcome, i.e., disease-free survival (DFS) and overall survival (OS) by stage, age, MDT, and period, were calculated using the Kaplan–Meier method. Analyses were performed using STATA 16.1 software (16.1, StataCorp LLC, College Station, TX, USA). In this study, we reported 95% confidence intervals (CI).

Since the case study also includes 2020, the year of the COVID-19 pandemic, and our province was strongly affected by the infection first and was affected by the restrictive measures of the so-called “I stay at home” decree, a table has been added with an additional figure to better explain the phenomenon.

## 3. Results

In the period 2013–2020, a total of 643 cases of endometrial cancer were registered (Table 1). Out of these cases, 319 were managed by an MDT (49.6%), while 324 were not (non-MDT, 50.4%). There are no differences in age between the two groups and no differences in age distribution. Among the 319 cases followed by an MDT, only 20% were diagnosed in 2013–2015, while the majority were diagnosed in recent years. Among the no-MDT cases, however, there are no large differences between the two periods. In terms of microscopic confirmations, there are no differences between the two groups, but the morphological characterization is better defined between MDT cases. With regard to stage, MDT cases are more often in stage I than non-MDT cases (83.7% vs. 62%) and have had more surgery (98.4% vs. 80.6%) and therapies (45.5% vs. 36.7%). However, no differences were underscored in terms of recurrences, which remained at approximately 10% in both groups (Table 1).

In the two study periods, the proportion of women managed by an MDT increased significantly, rising from 27.4% in 2013–2015 to 62.3% in 2016–2020 (Table 2). Notably, this increase was more pronounced among elderly women aged 65 and above, increasing from 22.3% to 64%. Moreover, women in districts of the metropolitan area, particularly those in mountainous regions such as Montecchio Emilia (increasing from 18.9% to 72.7%) and Castelnovo né Monti (rising from 25.0% to 69.7%), experienced a substantial surge in MDT consultations (Table 2). Over the years, there was a rise in the number of patients in stage I, increasing from 37.7% to 67.3%, who were managed by an MDT. Simultaneously, there was an increase in the number of women in stages II, III, and IV. Furthermore, in the more recent period, surgical interventions doubled (from 31.8% to 66%), and adjuvant therapies also significantly increased (from 24.4% to 69.7%). Although the overall number of recurrences remained relatively stable, occurring in 8% of women (19 out of 234) in the first period and 9% (37 out of 409) in the second period, there was a significant increase in the incidence of women experiencing recurrences who were subsequently managed by an MDT. This percentage rose considerably from 15.8%. to 75.7% in recent years.

Of the 56 relapses that occurred over the years (Table 3), 31 cases (55.4%) were monitored by the MDT panel. These patients, compared to no-MDT, were mostly over 65 (64.5% vs. 35.5%) and in stage I (65.6% vs. 34.4%), with recurrence within one year of diagnosis of endometrial cancer (78.6% vs. 21.4%).

The survival study (Table 4) showed an excess risk in terms of DFS for women in stage II [HR 2.47; 95% CI 1.27–4.81] and stage III [HR 3.24; 95% CI 1.86–5.65] for those aged over 65 [HR 4.42; 95% CI 1.59–12.28], and for patients having recurrence [HR 17.98; 95% CI 11.49–28.12]. Notably, patients who were not managed by an MDT exhibited only a slightly higher risk compared to those under MDT care [HR 2.10; 95% CI 0.75–1.61]. In terms of OS, the study confirmed an elevated risk for stage II [HR 3.68; 95% CI 1.74–7.78] and stage III patients [HR 4.02; 95% CI 2.00–8.08], as well as for those aged over 65 [HR 7.34; 95% CI 1.78–30.28]. Additional risk factors included not undergoing treatment [HR 1.75; 95% CI 1.00–3.07], experiencing relapse [HR 3.03; 95% CI 1.75–5.24], and not being managed by an MDT [HR 1.55; 95% CI 1.00–2.44].

Figure 1 shows significant variations in DFS by stage and age. Furthermore, women under MDT care had survival rates at 2, 4, 6, and 8 years (90%, 84%, 78%, and 64%) slightly higher than women without an MDT (86%, 77%, 69%, and 64%, respectively).

Figure 2 shows OS data, demonstrating that MDT-managed women had survival rates at 2, 4, 6, and 8 years (94%, 90%, 84%, and 72%) higher than those without the MDT support (89%, 80%, 75%, and 70%, respectively). Overall, the 5-year survival rate was 87% for MDT patients and 79% for those not managed by an MDT.

Finally, a note relating to COVID-19: throughout the period considered, we recorded a slight decrease in incidence and a slight increase in mortality, both of which were not significant (Appendix A). The decline in incidence occurred above all in 2020, when 68 cases were recorded, compared to 96 recorded in 2019 (Appendix A); however, the decline in diagnosed cases did not correspond to a decline in patients taken care of by the PDTA (54 in 2019 and 52 in 2020), respectively.

## 4. Discussion

The objective of this study was to assess whether patients diagnosed with EC and managed by an MDT demonstrate superior outcomes compared to those without MDT involvement. In addition, the study aimed to investigate any changes in the MDT approach over the years. Examining a consistent number of cases accrued in the RE-CR from 2013 to 2020, patients under the MDT approach, representing roughly half of the cohort, generally unveiled a more precise diagnostic definition. This was evidenced by a higher percentage of morphological characterization and staging, as well as a greater likelihood of undergoing both surgical procedures and adjuvant therapy. The incidence of relapses in both groups was around 10%, in line with the 11% reported in the literature [34]. However, MDT patients revealed distinct characteristics. They were frequently older and diagnosed at an earlier stage, and relapses were mainly intercepted within the first year after diagnosis. The percentage of early-stage recurrences was also quite comparable, standing around 66%. This phenomenon is probably linked with the centralized management of relapsed women, usually supervised by gynecologist oncologists in collaboration with medical and radiation oncologists, both within and outside our MDT. In these cases, the required procedures tend to be quite standardized, potentially contributing to the lack of differences in relapse identification and management between the MDT and non-MDT groups. However, our study showed a higher percentage of patients with relapse identified outside the MDT setting (45% in our study versus 33% in the American study). DFS did not show significant differences between the two groups, reflecting the fact that patient management does not change much for women without MDT consultation. Although they may not follow a dedicated path, these patients are still under the care of specialists, leading to comparable DFS outcomes. However, what did change was the management of relapses, as indicated by the excess OS risk (55%) noticed among the non-MDT groups compared to the MDT groups (HR 1.55, CI 1.00–2.44). Indeed, the pivotal role of an MDT in the comprehensive and effective management of patients, especially during the critical phase of relapse, is also underscored by the survival curves, which showed, in MDT-managed patients, higher values on average, which persisted even 5 years after diagnosis (OS rate 87% vs. 79%). Therefore, women under an MDT care generally experience better disease management and, presumably, effective handling of relapse. While MDT discussions may not completely overturn the prognosis of these patients, they play a crucial role in gynecological tumors. High-volume centers and MDT discussions are essential for adequate pre-treatment staging and adherence to guidelines. The ultimate aim is to minimize unnecessary treatments and increase survival [28]. Having dedicated specialists within the MDT is instrumental for a more appropriate diagnostic definition (e.g., employing the sentinel lymph node mapping technique) as well as for tailoring appropriate and effective treatment [29,30]. The collaborative expertise of specialized teams within the MDT framework significantly contributes to improved patient outcomes and overall quality of care. Undoubtedly, the COVID-19 emergency in recent years has introduced significant changes in EC patient management [35], influenced by factors such as lockdowns and the redirection of resources towards the management of COVID-19 patients. Thus, the importance of discussing individual cases within the context of the MDT was even more crucial [36]. The pandemic led to delays between MDT discussion and treatment, with the median duration increasing from 23 days in 2019 (pre-pandemic era) to 34 and 36 days in 2020 (during the pandemic with high restrictions) and 2021 (pandemic recovery period) [37]. Despite these challenges, adhering to guidelines within an MDT framework proved essential. This adherence not only ensured a better selection of female candidates for procedures like hysterectomy but also contributed to a cost reduction (by 41% due to decreased diagnostic tests) and mitigated possible adverse events [38]. This finding emphasizes the critical role of MDT in navigating the pandemic’s complexities, ensuring optimal patient care, and maximizing resource utilization. Even in patients with metastatic cancer, a discussion within an MDT seems to have contributed to reducing inappropriate treatments and minimizing the administration of chemotherapy during the last 14 days of life [39]. In our experience, the proportion of cases with metastatic cancer discussed within an MDT has increased over the years. Additionally, psychological assistance and the activation of palliative care, both in the hospital and in the local area, are available resources for these women. The presence of dedicated specialists within an MDT can better direct the patients’ path. For example, the presence of a geneticist on the team can assist in identifying tumors in women with hereditary breast–ovarian cancer syndrome or Lynch Syndrome. This involvement enables the identification of candidates for genetic testing of BRCA-1 and 2, activating surveillance programs [40]. These inherited mutations can heighten the risk of developing other cancers and affect the response to treatments [41]. At our center, patients requiring BRCA tests are identified, and tissue/blood samples are collected and sent to a centralized laboratory specializing in mutation research. If a mutation is detected, both patients and their relatives are subsequently referred to the same center to receive the necessary guidance and counseling. In recent years, precision oncology has increasingly developed in the treatment of EC. Therefore, the establishment of the molecular tumor board has become essential to improving clinical outcomes by personalizing treatment options based on vulnerabilities identified in the tumor genome. As part of our tumor board, the pathologist is supported by the molecular biologist, especially when dealing with rare molecular profiles. The presence of the pathologist within the MDT could facilitate the re-evaluation of conflicting reports, allowing a better characterization of the neoplasm and ensuring that the woman is directed toward the most appropriate treatment [42]. Moreover, our center maintains a dedicated gyneco-pathologist, ensuring consistent expertise. In cases of discrepancy, we have the option to send the slides to a pathologist who is a leading expert in the specific tumor type, further ensuring precise and reliable diagnoses. The management of an MDT appears to be more crucial than centralizing patients. An old Italian study concluded that centralization was neither feasible nor useful. Instead, it suggests that only selected high-risk patients should be referred to the reference centers [43]. A recent study conducted by our group highlighted the tasks of a high-volume center. These include making decisions about patients, referrals to smaller centers, performing histological diagnosis, selecting patients for adjuvant therapy, determining eligibility and treatment options for fertility-sparing approaches, and choosing appropriate follow-up protocols [44]. During MDT discussion, it is imperative to take into account not only the tumor characteristics but also the social conditions of the patient. This holistic approach ensures that patient care is not only medically sound but also tailored to the individual’s specific needs and social context, ultimately leading to improved outcomes and a higher quality of life. The evolution of the MDT approach over time is of significant interest. Introduced in our hospital in 2013, the MDT required a few years to reach its full potential. Initially, young women in their early stages and city residents primarily benefited from the MDT. However, over the years, not only has the number of patients under MDT care increased (from around 20 to 50 cases per year), but also the demographics of these patients have changed. In fact, there has been a notable rise in elderly women (with the average age increasing from 62.6 to 67.1 years) and patients diagnosed in advanced stages (from 8% to 49%) or locally advanced stages (from 8% to 96%). Moreover, there has been an increase in patients residing in hilly areas (e.g., from 18% to 65%) and mountainous regions (e.g., from 25% to 70%). The progressive growth in the number of patients managed by an MDT (from 27% to 62%) has corresponded with enhancements in both the surgical approach (from 32% to 67%) and the adjuvant treatment (from 24% to 70%). Among the strengths of this study, we must emphasize that the data are extracted from a cancer registry; therefore, there is no selection bias, and the data are from recent years. Unfortunately, among the limitations is the fact that the data refer to a single center; however, at the moment, the unavailability of recent data from other cancer registries in Italy does not allow multi-center work to be carried out.

## 5. Conclusions

In conclusion, the MDT approach appears to have a positive and lasting impact on EC patient survival. The MDT approach must provide for an overall management of the patient, which, in addition to surgery and treatments, also includes social and, last but not least, communicative aspects [45].

While patients cared for by an MDT do not experience fewer relapses than non-MDT patients, the management of relapses has most likely changed significantly. Finally, the study indicates that the MDT method evolves over time both quantitatively by expanding access to a larger number of women and qualitatively without bias in selecting women with the best prognosis. Overall, these findings emphasize the importance of an MDT in refining the approach to EC management.

## Figures and Tables

**Figure 1 healthcare-12-00064-f001:**
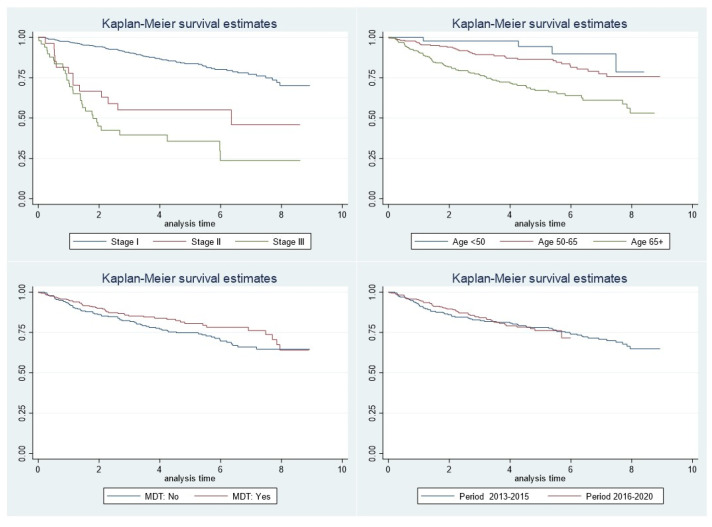
Reggio Emilia Cancer Registry, years 2013–2020. Kaplan–Meier curve of disease-free survival by stage, age, MDT, and period.

**Figure 2 healthcare-12-00064-f002:**
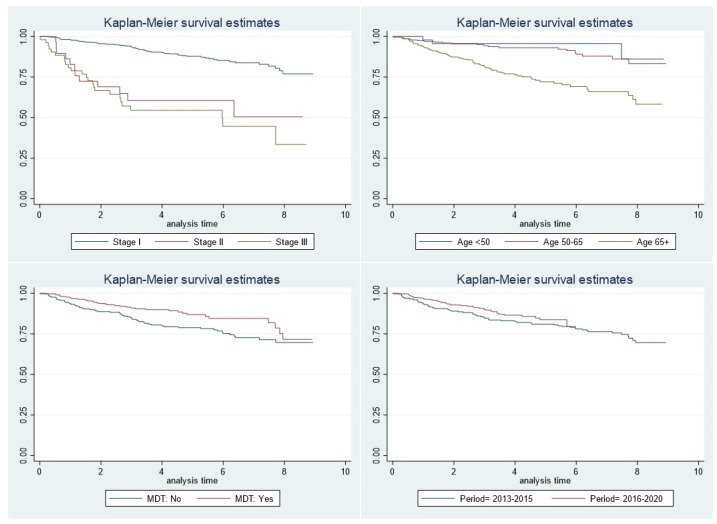
Reggio Emilia Cancer Registry, years 2013–2020. Kaplan–Meier curve of overall survival by stage, age, MDT, and period.

**Table 1 healthcare-12-00064-t001:** Reggio Emilia Cancer Registry, years 2013–2020. Characteristics of patients using MDT approach.

			MDT Yes		MDT No	
	n	%	n	%	n	%
**Overall**	643		319		324	
**Age at diagnosis**						
<50	54	8.4	20	6.3	34	10.5
50–65	252	39.2	130	40.7	122	37.7
65+	337	52.4	169	53.0	168	51.8
**Period of diagnosis**						
2013–2015	234	36.4	64	20.0	170	52.5
2016–2020	409	63.6	255	80.0	154	47.5
**Method of diagnosis**						
Histological	631	98.1	319	100	314	96.9
Clinical/instrumental	12	1.9	0	0.0	10	3.1
**Morphology**						
Adenocarcinomas	528	82.1	278	87.1	250	77.2
Cystic, mucinous, and serous neoplasms	44	6.8	23	7.2	21	6.5
Sarcoma	45	7.0	16	5.0	29	9.0
Others	26	4.0	2	0.6	24	7.4
**Stage**						
I	468	72.8	267	83.7	201	62.0
II	29	4.5	5	1.6	24	7.4
III	52	8.1	27	8.5	25	7.7
IV	46	7.2	17	5.3	29	9.0
Unknown	48	7.5	3	0.9	45	13.9
**Surgery**						
Yes	575	89.4	314	98.4	261	80.6
No	58	9.0	5	1.6	53	16.4
Unknown	10	1.6	0	0.0	10	3.1
**Therapies**						
Yes	264	41.1	145	45.5	119	36.7
No	349	54.3	172	53.9	177	54.6
Unknown	30	4.7	2	0.6	28	8.6
**Recurrence (stage I–III *)**						
Yes	56	10.2	31	10.3	25	10.0
No	463	84.3	249	83.3	214	85.6
Unknown	30	5.5	19	6.4	11	4.4

* Loco-regional recurrence and metastases were evaluated only for patients with stages I–III.

**Table 2 healthcare-12-00064-t002:** Reggio Emilia Cancer Registry, years 2013–2020. Distribution of MDT cases by age, district, stage, treatment, recurrence, and period.

	2013–2015			2016–2020		
	Total	MDT Yes		Total	MDT Yes	
	n	n	%	n	n	%
Overall	234	64	27.4	409	255	62.3
**Age at diagnosis, mean (SD)**	65.5 (12.6)	62.6 (10.8)		67.1 (11.9)	67.1 (11.0)	
**Age at diagnosis**						
<50	23	7	30.4	31	13	41.9
50–65	99	32	32.3	153	98	64.1
65+	112	25	22.3	225	144	64.0
**District**						
Correggio	26	3	11.5	42	19	45.2
Guastalla	24	1	4.2	64	24	37.5
Montecchio Emilia	37	7	18.9	44	32	72.7
Reggio Emilia	97	43	44.3	164	117	71.3
Scandiano	34	6	17.6	62	40	64.5
Castelnuovo ne’ monti	16	4	25.0	33	23	69.7
**Stage**						
I	162	61	37.7	306	206	67.3
II	13	0	0.0	16	5	31.3
III	26	2	7.7	26	25	96.2
IV	13	1	7.7	33	16	48.5
Unknown	20	0	0.0	28	3	10.7
**Surgery**						
Yes	201	64	31.8	374	250	66.8
No	24	0	0.0	34	5	14.7
Unknown	9	0	0.0	1	0	0.0
**Therapies**						
Yes	86	21	24.4	178	124	69.7
No	134	43	32.1	215	129	60.0
Unknown	14	0	0.0	16	2	12.5
**Recurrence (stage I–III *)**						
Yes	19	3	15.8	37	28	75.7
No	174	57	32.8	289	192	66.4
Unknown	8	3	37.5	22	16	72.7

* Loco-regional recurrence and metastasis were evaluated only for patients with stages I–III.

**Table 3 healthcare-12-00064-t003:** Reggio Emilia Cancer Registry, years 2013–2020. Distribution of 56 recurrences by age, stage, and time of recurrence with MDT approach.

	MDT			
	Yes		No	
	n	%	n	%
**Overall**	31	55.4	25	44.6
**Age at diagnosis**				
<50	0	0.0	3	100
50–65	11	50.0	11	50.0
65+	20	64.5	11	35.5
**Stage**				
I	21	65.6	11	34.4
II	3	42.9	4	57.1
III	7	41.2	10	58.8
**Recurrence (stages I–III)**				
<1 year	11	78.6	3	21.4
1–2 year	12	57.1	9	42.9
2+ years	8	38.1	13	61.9

**Table 4 healthcare-12-00064-t004:** Reggio Emilia Cancer Registry, years 2013–2020. Disease-free survival and overall survival by stage, age, therapies, MDT, and recurrence.

Characteristics	Disease-Free Survival		Overall Survival	
	HR	95% CI	HR	95% CI
**Stage**				
I	1.00	Ref.	1.00	Ref.
II	2.47	1.27–4.81	3.68	1.74–7.78
III	3.24	1.86–5.65	4.02	2.00–8.08
**Age at diagnosis**				
<50	1.00	Ref.	1.00	Ref.
50–65	1.60	0.56–4.57	2.18	0.50–9.42
65+	4.42	1.59–12.28	7.34	1.78–30.28
**Therapies**				
Yes	1.00	Ref.	1.00	Ref.
No	1.39	0.87–2.22	1.75	1.00–3.07
**MDT**				
Yes	1.00	Ref.	1.00	Ref.
No	1.10	0.75–1.61	1.55	1.00–2.44
**Recurrence (stages I–III)**				
No	1.00	Ref.	1.00	Ref.
Yes	17.98	11.49–28.12	3.03	1.75–5.24

## Data Availability

The data presented in this study are available on request from the corresponding author. The data are not publicly available due to ethical and privacy issues; requests for data must be approved by the Ethics Committee after the presentation of a study protocol.

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
