# Peer review of "Optimizing Outcomes through a Multidisciplinary Team Approach in Endometrial Cancer"

_healthcare, 2023, doi:10.3390/healthcare12010064_

Round 1

Reviewer 1 Report

Comments and Suggestions for Authors

In the submitted manuscript: " Optimizing Outcomes through a Multidisciplinary Team Approach in Endometrial Cancer", the authors presented a study assessing the impact of a multidisciplinary team (MDT) approach on the treatment outcomes of patients with endometrial cancer (EC), using data from 2013–2020 from the Reggio Emilia Cancer Registry.

The topic is interesting and important for the treatment of patients with Endometrial Cancer. There are several published works on this topic. New work presented in EMJ Oncol is also available. 2023;11[Suppl 4]:2-11 (DOI/10.33590/emjoncol/10309426. https://doi.org/10.33590/emjoncol/10309426.).

The authors of the work sent to Healthcare deal with this topic from the point of view of Reggio Emilia in 2013-2020. They presented the results in 4 tables and 2 figures.

Contributes data for 643 cases (2013-2020) in Reggio Emilia.

Is there no data for 2021 and 2022?

 Comparisons with other Italian regions could be made to make the work multi-centre.

Maybe a job for the future.

The results presented in the tables are clear, but chapter 3. Results is written chaotically.

Table 1 should be smaller and on one page (it doesn't read well).

There are probably errors (?) in Table 1 in "Recurrence (stage I-III). Please analyze and recalculate.

There are probably errors (?) in Table 2 in "Recurrence (stage I-III). Please analyze and recalculate.

Tables and Figures should be quoted first because it is easier to read and analyze the results.

Tables and Figures should be placed after the first citation because it is easier to read and analyze the result

Chapter 3 should be reworded, it is difficult to read.

Why Chapter 3.1. Figures and Tables? All this should be included when discussing the results.

Citations are appropriately selected. Of the 44 citations, 28 are from the last 5 years, including four from 2023.

Chapter 3 needs to be reworded.

Author Response

Thank you for your valuable comments, we hope we have responded clearly to your requests. Best regards, Lucia Mangone

Reviewer 2 Report

Comments and Suggestions for Authors

The article I reviewed carefully analyzed if the utilization of the MDT may affect the outcome of patients with EC who underwent surgical intervention between 2013-2020. This was an Italian single-center retrospective observational study based on pathological data, hospital discharge records and mortality data. Interestingly, women under MDT care were more likely to undergo surgical interventions and to receive more therapies. Survival rates /DFS and OS/ were better for MDT-managed women than for those not managed by MDT. The Authors stated that “…Comparing the 2013-2015 to 2016-2020 study periods, a shift towards caring for older women, more advanced-stage patients, and those residing outside the metropolitan area, along with a greater number of relapsed cases (from 16% to 76%) were accounted for.”.  

I feel this paper is worth publishing in the Healthcare Journal after minor revision.

1.     I would like to ask the Authors for clear definition of MDT-managed patients.

2.     I wonder did the Authors may forward some of the Tables into the supplementary material, if this changes did not decrease the power of the statistical data.

3.     Did the differences of MDT-managed and non MDT-managed EC patients existed between 2013-2019 and 2020, when the pandemic COVID-19 was present, especially in this region, with high restrictions?

4.     Check carefully the Reference list, some of the references should be gently corrected.  

Author Response

(The authors gave the same response as above.)
